# FINTRUTHQA: A BENCHMARK DATASET FOR EVALUATING THE QUALITY OF FINANCIAL INFORMATION DISCLOSURE

## ABSTRACT

Accurate and transparent financial information disclosure is crucial in the fields of accounting and finance, ensuring market efficiency and investor confidence. Among many information disclosure platforms, the Chinese stock exchanges' investor interactive platform provides a novel and interactive way for listed firms to disclose information of interest to investors through an online question-and-answer (Q&A) format. However, it is common for listed firms to respond to questions with limited or no substantive information, and automatically evaluating the quality of financial information disclosure on large amounts of Q&A pairs is challenging. This paper builds a benchmark FinTruthQA, that can evaluate advanced natural language processing (NLP) techniques for the automatic quality assessment of information disclosure in financial Q&A data. FinTruthQA comprises 6,000 real-world financial Q&A entries and each Q&A was manually annotated based on four key evaluation criteria: *question identification*, *question relevance*, *answer readability*, and *answer relevance*. We benchmarked various NLP techniques on FinTruthQA, including statistical machine learning models, pre-trained language model and their fine-tuned versions, as well as large language models (LLMs). Experiments showed that existing NLP models have strong predictive ability for *question identification* and *question relevance* tasks, but are suboptimal for *answer readability* and *answer relevance* tasks. By establishing this benchmark, we provide a robust foundation for the automatic evaluation of information disclosure, significantly enhancing the transparency and quality of financial reporting. FinTruthQA can be used by auditors, regulators, and financial analysts for real-time monitoring and data-driven decision-making, as well as by researchers for advanced studies in accounting and finance, ultimately fostering greater trust and efficiency in the financial markets.

## 1 INTRODUCTION

Within the financial market, enhancing transparency and timeliness in information disclosure can mitigate information friction, empower investors to make informed decisions, and foster market efficiency. According to Lee & Zhong (2022), retail investors struggle to integrate the existing public information, and increased online platform activities are associated with more active market transactions, such as greater trading volume, return volatility, and market liquidity. Novel investor interactive platforms (such as online forums established by the Shanghai Stock Exchange (SSE) and Shenzhen Stock Exchange (SZSE) ) offer a unique avenue for retail investors to engage directly with listed firms. In contrast to conventional modes of disclosure, such as mandatory disclosures and traditional voluntary disclosures like conference calls, which follow specific procedures and formats, these novel online platforms enable direct, interactive, near real-time communication between firms and ordinary investors through the format of Q&A. By analyzing the interactions on these platforms, researchers can gain insights into how effectively companies communicate with investors, the types of information that investors most value, and how the disclosure practices of listed firms may impact their corresponding trading patterns in financial markets. This will enhance financial market regulation and provide deeper insights into corporate disclosure practices.

Figure 1: Workflow of the financial Q&A quality evaluation process.

The interactive nature of these platforms provides a rich dataset for assessing the quality of information disclosure. However, the sheer volume of data and the diversity in the quality of questions and responses pose significant challenges for large-scale analysis. Leuz & Wysocki (2016) summarizes that the commonly used measures of disclosure in the financial sector primarily capture the quantity rather than the quality of the information provided. However, it is possible to construct the quality dimension of disclosure by focusing on specific disclosures, and Natural Language Processing (NLP) emerges as a promising solution to evaluate the quality of information disclosure at scale. Existing applications of NLP in automatic financial and accounting text analysis have demonstrated success in various domains, such as automatic analysis of annual financial reports(Smailović et al., 2017) and automatic sentiment analysis system on financial news articles(Van de Kauter et al., 2015). However, to the best of our knowledge, there is no standard dataset that evaluates the quality of financial information disclosure on these Q&A platforms.

In this work, we present a Q&A information disclosure quality assessment benchmark that evaluates Chinese Q&A data sourced from interactive platforms for communication between investors and listed companies, as established by SSE and SZSE. In developing this model, we have explored several technical challenges, including the interpretation of natural language, context sensitivity, and the subjective nature of relevance and readability assessments. This paper details the development, implementation, and evaluation of our NLP model, emphasizing its potential impact on the financial sector and the broader implications for automated quality assessment in investor communications. Datasets and code are available at `https://anonymous.4open.science/r/FinTruthQA-218F`.

## 2 RELATED WORK

**Disclosure measurement**  The measurement issue pertaining to disclosures has been a persistent challenge in empirical accounting research(Leuz & Wysocki, 2016). Predominantly, comprehensive measures have focused on quantifying disclosures, encompassing criteria such as the presence or absence of quarterly and annual reports, as well as numerical counts of activities like conference calls and earnings forecasts(Lee & Zhong, 2022). Another commonly used measure is the Association for Investment Management and Research (AIMR) rankings, based on annual surveys of financial analysts who evaluate U.S. firms' disclosure practices(Nagar et al., 2003; Healy et al., 1999; Welker, 1995). However, they are limited to large U.S. firms and cover only a specific time period. Furthermore, alternative measures designed for specific disclosures typically employ a keyword-based approach to analyze the tone and sentiment of well-established, traditional disclosures such as earnings forecasts and conference call transcripts(Bochkay et al., 2023; Frankel et al., 2010; Li, 2010).

**Financial NLP**  Financial documents, such as annual reports, financial news articles, and economic policy statements, are typically composed in technical language in a non-structured text format and possess unique features distinct from other domains.(Chang et al., 2016) Recent studies have focused on developing pre-trained models tailored for the financial domain (Araci, 2019; Yang et al., 2020; Liu et al., 2021), with a primary emphasis on the English language. These efforts are predominantly

Figure 2: Example of a Q&A pair and four quality evaluation tasks.

oriented towards sentiment classification tasks in the financial context (Wan et al., 2021). Additional research in this field has concentrated on question-answering tasks, involving datasets that combine textual and tabular information, where numerical analysis is often essential to derive answers (Zhu et al., 2021; Zhao et al., 2022; Li et al., 2022). To our knowledge, there has been no prior research or dataset dedicated to the automatic quality assessment of financial Q&A data.

# 3 DATASET CONSTRUCTION

## 3.1 DATA COLLECTION

The data for this study was obtained from the interactive platforms for communication between investors and listed companies established by the Shanghai Stock Exchange (Shangzhen E-Hudong[1]) and the Shenzhen Stock Exchange (Hudongyi [2]). These two stock exchange sites cover almost all listed companies in China. These Q&A interactions are authentic, generated by human users and company representatives, and are regulated by the China Securities Regulatory Commission, ensuring the accuracy of disclosures. The companies involved are legally obligated to ensure the accuracy of their disclosures, and failure to do so could result in legal consequences. This regulatory oversight ensures that the data we used reflects genuine interactions. We collected over 900,000 Q&A entries, among which we randomly selected 6,000 samples for manual annotation, with 3,000 entries from each exchange's platform. The data collection involved using web scraping techniques to extract the Q&A entries from the official platforms. The data were then manually reviewed and processed to ensure quality and consistency. The process was overseen by a PhD student from the Business School of XXX University. The ethics statement, including considerations of general ethical conduct and potential negative societal impacts, can be found in Appendix Section A. We also give several examples in Appendix Section G to demonstrate the wide range of investor inquiries covered in our dataset.

## 3.2 DATA PRE-PROCESSING

We found that the Shanghai Stock Exchange (SSE) data includes the company name and stock code at the beginning of the question texts, while the Shenzhen Stock Exchange (SZSE) data does not. To assess the impact of this discrepancy, we conducted experiments comparing Q&A pairs with and without the company name as a prefix and found negligible differences. Therefore, we removed these prefixes from the SSE data in our experiments.

---

[1] https://sns.sseinfo.com/
[2] https://irm.cninfo.com.cn/

## 3.3 DATA ANNOTATION

We focus on four key information disclosure quality evaluation criteria[3]: *question identification*, *question relevance*, *answer relevance*, and *answer readability*. These criteria are widely recognized as crucial indicators of information quality and are important for investors to consider when evaluating Q&A information. Specifically, the criteria in FinTruthQA are defined as follows:

1. **Question identification**: This criterion assesses whether the question from an investor is a genuine question, as opposed to a suggestion or complaint. Each question is assigned a binary label: *Positive*, indicating it is a real question, or *Negative*, indicating it is not. Data entries with questions labeled as *Negative* are excluded from further evaluation under the remaining criteria.

2. **Question relevance**: This criterion evaluates whether a question is pertinent to the investment topic, labeling it as *Positive* if relevant and *Negative* if not. It assesses whether the question is related to the company's shareholders, financial indicators, industry conditions, or other investment-related topics.

3. **Answer readability**: This criterion evaluates how easily an answer can be understood and whether it is well-written. The readability score ranges from 1 to 4, with 4 indicating the highest level of readability—clear, concise, and free of ambiguity. A score of 3 indicates moderate readability with some issues, while a score of 2 reflects significant readability challenges. A score of 1 suggests the answer is unreadable, with major ambiguities, incoherent sentences, or content unrelated to the question.

4. **Answer relevance**: This criterion evaluates whether an answer is relevant and informative in response to the question. It assesses whether the answer provides sufficient detail and explanation to address the question. Answers are classified into three levels: Level 1 indicates the answer is completely unrelated to the question (none of the questions is answered); Level 2 indicates the answer is partially related (some questions are answered or it is difficult to judge) ; and Level 3 indicates the answer fully responds to the question (all questions are answered).

The importance of these criteria lies in their collective ability to enhance the quality and reliability of information on investor communication platforms. The annotation process was carried out in three phases: pre-annotation, formal annotation, and data verification and integration. In the pre-annotation phase, we recruited six students from the Accounting Department at XXX University and conducted five rounds of annotation. After the last round, the inter-annotator agreement (IAA) rate was calculated, and the four students with the highest average IAA rates (85.52%)—two undergraduates and two master's students—were selected for the formal annotation phase. To ensure annotation quality, the four selected annotators were evenly divided into two groups, with each group annotating the same data instances. Discrepancies were first discussed and resolved within the group, with unresolved issues escalated to the project leader for final decisions. After the formal annotation, the project leader verified the data to ensure alignment with the latest annotation criteria and completeness, before integrating the data and summarising key information, such as sample size and score distribution. Figure 2 illustrates an example of the annotation. A detailed description of the annotation process is listed in the Appendix Section B.

## 3.4 QUALITY CONTROL

For the *answer relevance* task, we observed that the raw annotations contained noise due to the subjective nature of this task. To address this, we introduced an additional quality control step to correct the raw annotations. We implemented an enhanced correction process inspired by Confident Learning (Northcutt et al., 2021a) principles. Sample data points demonstrating inaccuracies, along with implementation details, can be found in Appendix Section C.

---

[3]In this paper, the terms "criteria" and "tasks" are used interchangeably. "Criteria" refers to the specific standards used to evaluate the quality of the Q&A pairs, while "tasks" refers to the individual evaluation processes performed based on these criteria, particularly in the experimental sections.

Table 1: Dataset Statistic of Different Tasks.

| Task | Label | # Train | # Valid | # Test | # Label | Freq. | Total |
|---|---|---|---|---|---|---|---|
| Question identification | Negative | 327 | 90 | 81 | 498 | 8.30% | 6,000 |
| | Positive | 3,673 | 910 | 919 | 5,502 | 91.70% | |
| Question relevance | Negative | 714 | 6 | 7 | 27 | 0.49% | 5,502 |
| | Positive | 3,654 | 911 | 910 | 5,475 | 99.51% | |
| Answer readability | 1 | 106 | 190 | 198 | 1,094 | 19.88% | 5,502 |
| | 2 | 26 | 3 | 6 | 35 | 0.64% | |
| | 3 | 68 | 19 | 20 | 107 | 1.94% | |
| | 4 | 2,868 | 705 | 693 | 4,266 | 77.54% | |
| Answer relevance | 1 | 731 | 194 | 206 | 1,131 | 20.63% | 5502 |
| | 2 | 533 | 136 | 130 | 779 | 14.21% | |
| | 3 | 2,404 | 587 | 581 | 3,572 | 65.16% | |

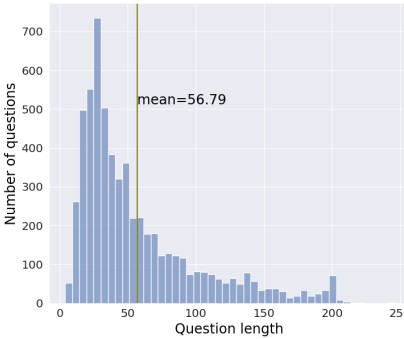 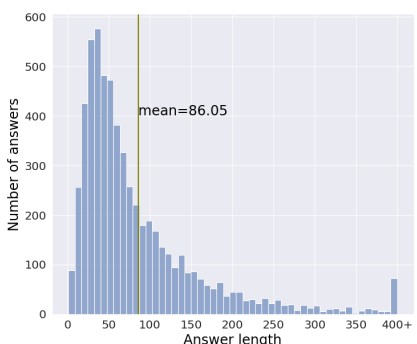

Figure 3: Length distributions of questions and answers (in characters).

### 3.5 DATASET STATISTICS

The complete annotated dataset comprises 6,000 samples labeled for *question identification*, *question relevance*, *answer relevance*, and *answer readability* tasks, with each data point having four distinct labels corresponding to these tasks. The dataset was randomly divided into training, validation, and testing sets in a ratio of 4:1:1. The label distributions for each task are detailed in Tables 1. The provided statistics reflect the data after undergoing the quality control process. Note that our later analysis focuses exclusively on real questions. Only for the first task (*question identification*), we used the full set of 6,000 samples. For the remaining three tasks, we removed non-question samples, as they could skew the model's understanding and performance, especially for tasks like *question relevance*, where the context of a real question is crucial for accurate assessment. It is worth noticing that the *question relevance* labels are highly imbalanced, with only 6 and 7 negative labels in the validation and test sets, respectively. We retained this task as the imbalance reflects the real-world data distribution, and the concept of *question relevance* is important in financial information disclosure. Figure 3 shows the length distribution of questions and answers, measured in characters, for all 6,000 QA pairs.

## 4 BENCHMARKS

### 4.1 MODEL SELECTION

To fully evaluate the performance of different methods in FinTruthQA, we benchmarked several models that cover Machine Learning (ML)-based, Pre-trained Language Model (PLM)-based, PLM with continued Pre-Training (CPT)-based, and LLM-based approaches. Given the distinct nature and varying difficulty of the tasks, we applied tailored approaches to each accordingly.

**ML-based.** We modeled these tasks as text classification problems and employed various advanced machine learning models, including Logistic Regression (LR)(Hosmer Jr et al., 2013), Support Vector Machine (SVM)(Hearst et al., 1998), K-nearest neighbor (KNN)(Peterson, 2009), and Random Forest (RF)(Breiman, 2001). For semantic feature extraction, we first utilized the Jieba toolkit

[4] for Chinese word segmentation and then extracted weighted text representations using TF-IDF Sparck Jones (1972). To achieve optimal performance, we used cross-validation and GridSearch to rigorously explore the best hyperparameters. The best-performing classifiers and hyperparameters varied across tasks, reflecting the distinct nature of each task. As a result, we performed task-specific hyperparameter searches for each task individually.

**PLM-based.** In our experiments, we found that the tasks of *question identification* and *question relevance* were relatively straightforward, with BERT achieving excellent performance on these tasks. Therefore, our focus shifted to exploring the effectiveness of PLM-based methods on the more challenging tasks of *answer relevance* and *answer readability*. Specifically, we investigated a range of BERT-based models, including those specifically trained on financial text data, such as financial news, research reports, and financial encyclopedia entries (*FinBERT,*[5] *Mengzi-fin* (Zhang et al., 2021), and *DKPLM* (Zhang et al., 2022)). Furthermore, we examined whether fine-tuned PLMs on specific tasks, such as NLI (Natural Language Inference) and QA (Question Answering), would offer any benefits for these tasks (*SBERT-nli* (Reimers & Gurevych, 2019) and *RoBERTa-extractive-qa* (Liu et al., 2019)). Finally, we also examined the performance of a large BERT model (*BERT (Large)*).[6] Notably, unless otherwise specified, all PLMs employed in this study are the base and Chinese versions. Links for these PLMs can be found in Appendix Section D.

**PLM with CPT-based.** General domain PLMs, trained on broad texts, often lack the specialized knowledge required to handle the nuanced language and terminology in finance. This limitation is especially pronounced in tasks like *answer relevance*, where understanding domain-specific jargon and abbreviations is essential for accurate predictions. To bridge this gap, we curated a substantial volume of domain-specific texts and conducted continued pre-training on general PLMs. Specifically, we collected over 900,000 Q&A pairs from the SSE and SZSE platforms by web scraping. The top-performing models from previous experiments were then selected for continued pre-training using masked language modeling (MLM). This pre-training on a large-scale, domain-specific dataset enhances the PLM's ability to understand financial contexts, capture more specialized information, and achieve better performance in downstream tasks.

**LLM-based.** We further evaluate some advanced LLMs, including GPT-4 (Achiam et al., 2023), LLaMA-3.1 (Dubey et al., 2024), Qwen2 (Yang et al., 2024), and Mistral (Jiang et al., 2023), on the most challenging *answer relevance* task. Due to their in-context learning ability, LLMs can perform this task in a zero-shot manner, without requiring specific fine-tuning. Specifically, we provided LLMs with task descriptions and evaluation criteria, instructing them to directly output relevance scores (*Direct score*). Moreover, we incorporated the Chain of Thought (CoT) (Wei et al., 2022) technique, prompting the LLMs to provide specific reasoning along with the relevance scores (*Score with CoT*). Full prompts and details are provided in Appendix Section F.

## 4.2 SETTINGS AND METRICS

**Implementation details.** For the ML-based models, we utilized the standard `scikit-learn`[7] library. For all PLM-based methods, we used `PyTorch`[8] and HuggingFace's framework[9] for downstream fine-tuning and evaluation. Detailed hyperparameter settings for our PLM-based experiments can be found in Appendix Section E. The tasks of *question identification* and *question relevance* only required encoding the questions as input, while *answer readability* and *answer relevance* required encoding both the question and the answer. The sequence started with a special classification token [CLS], and the question and answer were separated by the [SEP] token . During continued pre-training, we added two special tokens, [Q] and [A], into the vocabulary and prefixed them to the question and answer, respectively. This helps the PLMs better identify the components of each sentence. The continued pre-training was performed based on the UER framework (Zhao et al., 2019). All tests of LLMs were conducted using the transformer framework[10], except for the GPT-4 experi-

---

[4] https://github.com/fxsjy/jieba
[5] https://github.com/valuesimplex/FinBERT
[6] https://huggingface.co/yechen/bert-large-chinese
[7] https://scikit-learn.org/stable/
[8] https://github.com/pytorch/pytorch
[9] https://github.com/HuggingFace
[10] https://github.com/huggingface/transformers

Table 2: Model evaluation results of *question identification* and *question relevance* task.

| Category | Model | Question identification (%) | | | | Question relevance(%) | | | |
|---|---|---|---|---|---|---|---|---|---|
| | | Accuracy | Precision | Recall | F1 | Accuracy | Precision | Recall | F1 |
| ML-based | TF-IDF + LR | 93.70±0.86 | 94.11±0.75 | 99.35±0.18 | 96.66±0.48 | 99.42±0.13 | 99.42±0.13 | **100.00**±0.00 | 99.71±0.07 |
| | TF-IDF + RF | 93.70±0.65 | 94.02±0.63 | 99.46±0.18 | 96.66±0.37 | 99.42±0.13 | 99.42±0.13 | 99.93±0.10 | 99.67±0.08 |
| | TF-IDF + SVM | 93.60±0.73 | 93.78±0.76 | **99.64**±0.13 | 96.62±0.40 | 99.42±0.13 | 99.42±0.13 | **100.00**±0.00 | 99.71±0.07 |
| | TF-IDF + KNN | 92.67±1.11 | 93.27±0.98 | 99.16±0.37 | 96.12±0.61 | 99.42±0.13 | 99.42±0.13 | **100.00**±0.00 | 99.71±0.07 |
| PLM-based | BERT | **96.89**±0.55 | **97.64**±0.22 | 99.02±0.41 | **98.32**±0.28 | **99.67**±0.11 | **99.72**±0.12 | 99.96±0.05 | **99.85**±0.06 |

Table 3: Model evaluation results of *answer readability* task.

| Category | Model | F1 (Micro) | F1 (Macro) | F1 (Weighted) | QWK |
|---|---|---|---|---|---|
| ML-based | TF-IDF + LR | **81.72** ±0.27 | **34.91**±0.22 | **78.90**±0.27 | **41.20**±0.80 |
| | TF-IDF + RandomForest | 80.48±0.70 | 32.81±1.06 | 76.85±1.03 | 33.79±3.87 |
| | TF-IDF + SVM | 81.64±0.37 | 33.92±0.38 | 78.15±0.40 | 37.98±1.48 |
| | TF-IDF + KNN | 78.99±0.34 | 31.07±1.23 | 74.95 ±0.84 | 27.31±4.28 |
| PLM-based | BERT | 86.04±0.44 | 41.97±0.67 | 84.89±0.44 | 63.52±1.43 |
| | RoBERTa | 86.26±0.62 | 42.02±1.92 | 85.08±0.89 | 64.00±0.96 |
| | FinBERT | **87.57**±0.39 | 41.34±0.45 | **85.98**±0.69 | **66.53**±1.83 |
| | Mengzi-fin | 86.99±0.80 | 41.93±1.42 | 85.77±1.20 | 66.33±1.81 |
| | DKPLM | 87.24±0.76 | 41.90±1.17 | 85.76±0.72 | 65.78±0.63 |
| | SBERT-nli | 85.31±1.05 | 39.28±0.81 | 83.48±1.38 | 58.19±2.77 |
| | RoBERTa-extractive-qa | 87.21±0.52 | 41.22±0.11 | 85.75±0.64 | 65.97±0.80 |
| | BERT (Large) | 87.24±1.03 | 42.17±1.66 | 85.78±1.30 | 65.92±2.74 |
| | RoBERTa (Large) | 86.48±1.48 | **45.51**±4.59 | 85.37±1.41 | 64.58±2.11 |

ments, which were conducted by calling OpenAI's API with version GPT-4-0613. Specifically, we used Llama-3.1 with 8 billion parameters, and both Qwen2 and Mistral with 7 billion parameters.

**Metrics** For *Question Identification* and *Question Relevance*, which are binary classification tasks, we used accuracy, precision, recall, and F1-score as evaluation metrics. For *Answer Readability* and *Answer Relevance*, which are multi-class classification tasks, we calculated both micro and macro F1-scores, along with the Quadratic Weighted Kappa (QWK) (Dong et al., 2017), which is well-suited for ordinal classification by penalizing misclassifications based on their distance from the true label. Each model was independently run five times, and the mean and variance of the results are reported.

## 4.3 RESULTS AND ANALYSIS

### 4.3.1 TASK 1: QUESTION IDENTIFICATION

In this task, we benchmarked several ML-based and PLM-based models, with their performances summarized in Table 2. The ML-based models showed consistent results with minor variations. Both TF-IDF + LR and TF-IDF + RF reached an accuracy of 93.70% and an F1 score of 96.66, with closely aligned precision and recall. TF-IDF + SVM exhibited slightly lower accuracy but achieved the highest recall among the ML models, demonstrating its strength in identifying relevant instances. TF-IDF + KNN, while performing well with an accuracy of 92.67%, had the lowest precision and F1 score in this group. On the other hand, the PLM-based BERT model outperformed all ML-based methods, achieving the highest accuracy, precision, and F1 score. This result highlights the PLM model's ability to capture deeper contextual nuances, which traditional frequency-based methods like TF-IDF struggle to achieve.

### 4.3.2 TASK 2: QUESTION RELEVANCE

Table 2 presents the evaluation results for the question relevance task. The PLM-based model out-performed the ML-based models on all metrics except recall, achieving an accuracy of 99.67%, an F1 score of 99.85%, precision of 99.72%, and recall of 99.96%. This demonstrates its ability to effectively identify relevant questions while maintaining a strong balance in correctly classifying rare negative samples. In contrast, the TF-IDF-based models showed nearly identical performance across metrics, primarily predicting all samples as positive. Given the highly imbalanced label distribution (910 positive vs. 7 negative samples in the test set), these models achieve high recall by predicting positives but struggle to correctly identify the few negative instances, reflecting the challenge posed by the skewed data.

Table 4: Model evaluation results of *answer relevance* task.

| Category | Model | F1 (Micro) | F1 (Macro) | F1 (Weighted) | QWK |
|---|---|---|---|---|---|
| ML-based | TF-IDF + LR | 69.61±0.49 | **49.74**±0.53 | **65.07**±0.60 | **38.16**±2.13 |
| | TF-IDF + RF | 69.97±0.34 | 44.79± 0.84 | 63.07±0.76 | 36.63±1.10 |
| | TF-IDF + SVM | **70.08**±0.83 | 44.42±0.43 | 62.93±1.07 | 36.78±1.65 |
| | TF-IDF + KNN | 66.66±1.03 | 38.63±2.05 | 57.87±1.42 | 23.22±3.36 |
| PLM-based | BERT | 77.90±0.98 | 67.64±0.94 | 76.79±0.75 | 63.95±0.85 |
| | RoBERTa | 77.10±1.34 | 65.95±2.64 | 75.77±1.35 | 62.22±2.46 |
| | FinBERT | 78.19±0.96 | 68.18±0.79 | 77.25±0.81 | 65.49±0.54 |
| | Mengzi-fin | 78.04±0.72 | **68.96**±2.42 | **77.44**±0.82 | **66.31**±1.10 |
| | DKPLM | 76.92±0.57 | 63.00± 3.72 | 74.65±1.15 | 62.82±1.78 |
| | SBERT-nli | 74.52±0.36 | 61.18±2.54 | 72.38±1.07 | 53.88±1.12 |
| | RoBERTa-extractive-qa | **78.33**±0.36 | 67.02±1.48 | 77.12±0.69 | 64.66±1.44 |
| | BERT (Large) | 78.01±0.78 | 65.94±1.62 | 76.16±0.56 | 63.53±0.90 |
| | RoBERTa (Large) | 78.30±0.94 | 68.50±2.09 | 77.40±1.54 | 64.52±1.56 |
| PLM with CPT-based | FinBERT | **80.04**±0.23 | **70.58**±0.90 | **79.02**±0.18 | **67.89**±1.26 |
| | RoBERTa-extractive-qa | 78.34±0.34 | 68.01± 1.71 | 77.45±0.74 | 66.76±0.58 |
| | BERT (Large) | 78.12 ±0.65 | 65.15±3.97 | 76.02± 1.16 | 64.82±2.07 |
| | RoBERTa (Large) | 74.37±1.78 | 61.40±1.28 | 72.46±0.73 | 53.91±3.42 |
| LLM-based | GPT-4 (Direct score) | **67.39** | **63.11** | **70.14** | **59.06** |
| | GPT-4 (with CoT) | 59.76 | 58.45 | 64.28 | 57.51 |
| | Llama-3.1 (Direct score) | 53.00 | 41.59 | 56.13 | 43.24 |
| | Llama-3.1 (with CoT) | 46.46 | 40.82 | 51.57 | 39.56 |
| | Qwen2 (Direct score) | 60.52 | 31.33 | 52.83 | 23.68 |
| | Qwen2 (with CoT) | 59.00 | 38.97 | 57.36 | 42.35 |
| | Mistral (Direct score) | 54.53 | 30.40 | 50.99 | 29.57 |
| | Mistral (with CoT) | 59.32 | 23.03 | 60.20 | 38.90 |

### 4.3.3 TASK 3: ANSWER READABILITY

The evaluation results for the task *answer readability* are shown in Table 3. Among the ML-based models, the LR classifier exhibited the highest performance, achieving an F1 Micro score of 81.72% and a QWK of 41.20%. While LR provided reasonable performance, the relatively low F1 Macro score highlights its difficulties in handling class imbalance effectively. In contrast, the PLM-based models demonstrated significant improvements over the ML-based models. For instance, BERT achieved an F1 Micro score of 86.04% and a QWK of 63.52%, showcasing its superior ability to capture the complexities of readability assessment through richer contextual understanding.

**Detailed Evaluation of FinBERT's Performance.**    FinBERT outperformed all other models with an F1 Micro score of 87.57%, an F1 Macro score of 41.34%, an F1 Weighted score of 85.98%, and the highest QWK of 66.53%. Its pre-training on financial text provided a distinct advantage for this domain-specific task, enabling it to achieve the highest accuracy and alignment with human judgments. To further analyze FinBERT's performance, we visualize its confusion matrix in Figure 4a. We find that the model performs well in recognizing answers with the highest readability, but it struggles to accurately differentiate between the lower readability levels, highlighting the need for improvement in capturing finer distinctions in readability.

### 4.3.4 TASK 4: ANSWER RELEVANCE

**PLM-based Methods Consistently Outperform ML-based ones.**    As shown in Table 4, all PLM-based methods can achieve better performance than ML-based methods across all metrics. For instance, RoBERTa-extractive-qa achieved the highest Micro F1 score of 78.33%, while Mengzi-fin excelled with the highest Macro F1 score of 68.96% and a strong QWK of 66.31%. Their superior performance can be attributed to pre-training on task-specific and domain-specific data: RoBERTa-extractive-qa was pre-trained on QA datasets, which enhanced its ability to comprehend and evaluate answer relevance, while Mengzi-fin was pre-trained on financial corpora, enabling it to better capture domain-specific nuances in the financial context.

**Impact of Continued Pre-training on PLM-based Methods.**    We selected four PLM-based models and further pre-trained them using 900,000 Q&A pairs. As shown in Table 4, we observed that only FinBERT demonstrated significant performance improvements, achieving the highest Micro F1 score of 80.04% and a QWK of 67.89%, indicating its enhanced ability to assess answer relevance in financial contexts. In contrast, the other three models (RoBERTa-extractive-qa, BERT Large, and RoBERTa Large) did not exhibit consistent improvements. This suggests that domain alignment plays a crucial role in the effectiveness of continued pre-training, as FinBERT's domain-specific

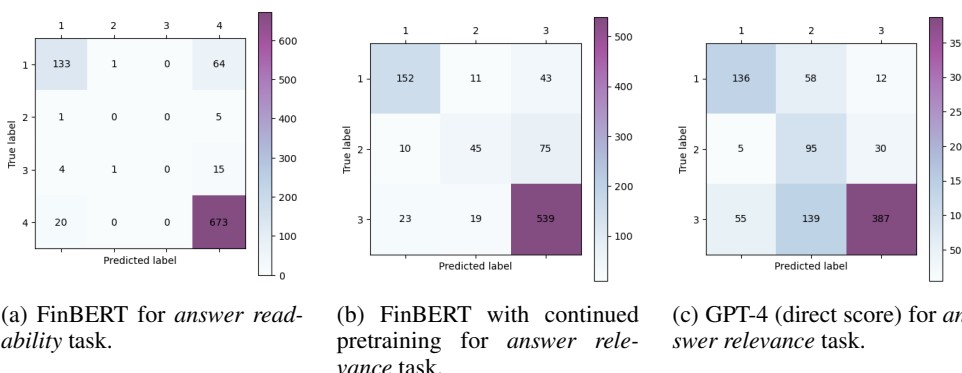

(a) FinBERT for *answer read-ability* task.

(b) FinBERT with continued pretraining for *answer relevance* task.

(c) GPT-4 (direct score) for *answer relevance* task.

Figure 4: Confusion matrix for models in Task 3 Answer readability and Task 4 Answer relevance.

pre-training on financial texts provided a strong foundation, whereas the other models may have experienced a mismatch between their learned representations and the specific requirements of the task, resulting in less notable gains.

**Suboptimal Performance of LLMs in Chinese Financial Contexts.** As shown in Table 4, we also explored the performance of some representative LLMs on answer relevance task. We observe that GPT-4 (Direct score) consistently outperform other LLMs but still lagged behind supervised PLM-based models, underscoring the difficulty LLMs face in understanding Chinese financial disclosures. Notably, GPT-4 with CoT did not outperform GPT-4 Direct score, suggesting that the CoT mechanism may introduce confusion rather than improving performance in this task. A potential reason is that GPT-4 was primarily trained on English corpora; it lacks sufficient financial domain knowledge in Chinese. This may result in knowledge gaps and hallucinations during the CoT process, thereby reducing the model's performance. Another study also observed a similar phenomenon where CoT failed in healthcare domain prediction with Chinese text. (Liu et al., 2024) We also tested Chinese financial LLMs, including CFGPT2(Li et al., 2023) and DISC-FinLLM(Chen et al., 2023), but found that these models could hardly follow instructions and failed to generate reasonable predictions. As a result, we decided to exclude them from our study. We anticipate that further domain adaptation methods, such as few-shot learning or retrieval-augmented generation (Lewis et al., 2020), may improve the performance of LLMs on this task. Figure 4b shows the confusion matrix for the best-performing model (FinBERT with continued pretraining). It demonstrates that the model effectively captures answers that are completely related to the questions (labeled as 3). However, the model struggles to distinguish partially unrelated answers (labeled as 2) from completely related ones, where the nuanced differences are subtle and challenging. The confusion matrices for GPT-4 (direct score) are shown in Figures 4c, which illustrate that the model tends to mix up answers that are completely irrelevant (labeled as 1) with those that are partially relevant (labeled as 2). Also, there's a tendency for the model to predict label 2.

**Error Analysis on GPT-4 Performance.** We have also conducted a thorough examination of bad cases to better understand the prevalent error types exhibited by GPT-4 in relevance scoring. One common error we identified occurs when the model generates an answer that revolves around the key terms of the query but fails to address the core question, leading to an incorrect judgment of relevance where there is none. Additionally, issues with pronoun resolution have led to errors where the question concerns one subject, such as employee benefits, but the answer discusses another, such as salaries, resulting in a misjudgment of irrelevance. Another notable issue is the model's handling of answers that state "the situation is unclear." In such cases, GPT-4 often misclassifies these as unclear responses, rather than recognizing them as valid.

## 5 LIMITATIONS

There are several limitations in this study. First, the Chinese cultural and regulatory context of SSE and SZSE may limit the generalizability of the findings to other markets. Future research is needed

Table 5: Examples of GPT-4 (direct score) bad cases and error analysis.

| Question | Answer | Prediction | Label | Error analysis |
|---|---|---|---|---|
| 请问贵公司2020年丰水季节弃电量是多少？

Could you please tell me what the amount of curtailed electricity was for your company during the 2020 flood season? | 尊敬的投资者您好！公司丰水期弃水电量受当年气候、来水量、网络负荷、相关政策等综合影响。谢谢关注。
Dear investors, hello! The amount of abandoned hydroelectric power during the company's flood season is influenced by various factors, including climate, water inflow, network load, and relevant policies. Thank you for your attention. | 2 | 1 | Model focuses on matching keywords instead of understanding the core intent of the question. |
| 董秘你好，贵公司今年利润下降，有没有考虑过降低员工待遇或者取消年终奖来维持利润

Hello, Company Secretary. Given your company's decline in profits this year, have you considered reducing employee benefits or canceling year-end bonuses to maintain profits? | 您好！公司注重提升员工的幸福感，在2017年初，公司普遍上调员工工资。对于年终奖，公司将综合考虑员工考核结果及公司业绩等多方面因素来确定。
Hello! The company places great emphasis on enhancing employee well-being. At the beginning of 2017, the company generally increased employee salaries. For the year-end bonus, the company will consider various factors, including employee performance evaluations and overall company performance, to determine the amount. | 2 | 3 | Error in pronoun resolution (e.g., salaries vs. employee benefits), causing confusion between related subjects. |
| 金风科技投资温州深远海海上风电项目拟引进塔筒，机舱罩等配套企业，公司是否参与
The investment by Jinfeng Technology in the Wenzhou Deep Sea Offshore Wind Power Project plans to introduce supporting enterprises for tower tubes, nacelle covers, etc. Is the company involved? | 尊敬的投资者，该情况尚不明确。感谢您的建议。

Dear Investor, the situation is still unclear. Thank you for your suggestion. | 2 | 3 | Model fails to interpret "unclear" as a valid state rather than an irrelevant response. |

to explore the applicability of the developed models and methodologies in different contexts and validate the findings across diverse settings. Second, the annotation process inherently involves a degree of subjectivity, as differences in annotators' interpretations can introduce variability in the labeled data. To mitigate this, we conducted multiple rounds of pre-annotation and formal annotation with professional annotators, along with thorough error analysis and expert reviews to ensure consistency. Third, while precision, recall, F1-score, and quadratic weighted kappa (QWK) are standard metrics for evaluating model performance, they may not fully capture the qualitative aspects of information disclosure quality. The reliance on these metrics could lead to an overemphasis on quantitative evaluation while neglecting interpretability. Lastly, the textitquestion relevance task suffers from a highly imbalanced dataset, which may limit the model's ability to effectively learn from and evaluate rare instances. However, we chose to retain this task in the benchmark to reflect the real-world data distribution in financial information disclosure.

## 6 CONCLUSION

In this study, we introduce FinTruthQA, a benchmark designed to evaluate advanced NLP techniques for the automatic quality assessment of information disclosure in financial Q&A data. We constructed a dataset of 6,000 real-world financial Q&A entries, each manually annotated based on four key evaluation criteria of accounting. Experiments revealed the effectiveness of pre-trained BERT models in tackling these tasks. In particular, we observed that models pre-trained on financial corpora, as well as those designed for related tasks such as QA, outperformed other variants of BERT and GPT-4. This emphasized the importance of domain-specific and task-oriented training in improving performance. Although existing NLP models showed reasonable performance on this benchmark, there is still significant room for improvement, especially in the task of *answer relevance*, which calls for better NLP models. Our benchmark is not only useful for post-hoc regulatory assessments but also holds significant value in real-time applications. For instance, auditors, financial analysts, and corporate communication teams can leverage our framework for ongoing monitoring and assessment of information disclosure quality. This real-time application enhances market efficiency by enabling quicker identification and correction of inadequate disclosures, thereby benefiting all market participants, including retail investors. Moreover, our benchmark can serve as a valuable resource for academic research in natural language processing, particularly in areas concerning the automated assessment of text quality.

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

## A ETHICS

**General ethical conduct.** Our research on assessing the quality of information disclosure in financial Q&A using BERT-based models utilizes publicly available data from the Shanghai Stock Exchange and the Shenzhen Stock Exchange's interactive platforms. Following ethical guidelines, we ensured all data was anonymized to protect individual identities. The dataset consists of Q&A entries that are openly displayed on the interactive platforms and are used in compliance with the respective exchanges' policies and the China Securities Regulatory Commission's guidelines. To ensure transparency and reproducibility, we provide the code and pre-trained models used in our experiments, subject to any legal or ethical restrictions. To mitigate algorithmic bias, our annotation process included multiple rounds of pre-annotation and formal annotation involving professional annotators, ensuring a diverse and representative sample of Q&A data. Rigorous data cleaning and error analysis were conducted to address biases. We encourage users to perform their own checks to avoid potential biases related to specific sub-populations or other contextual factors.

**Potential negative societal impacts.** The primary goal is to develop models for evaluating disclosure quality rather than conducting direct causal analyses. Users should be cautious when applying these models in contexts other than financial Q&A, as the models may have learned information specific to the financial domain and Chinese market settings. Additionally, over-reliance on automated systems can lead to the neglect of human judgment and expertise. We recommend using our models as supplementary tools to assist human analysts rather than replace them entirely. Although our work aims to improve transparency and information quality in financial markets, users should take appropriate precautions to avoid any potential negative societal impacts resulting from misapplication of the models.

## B ANNOTATION PROCESS

The guideline of the general annotation process is depicted in Figure 5. The annotation process consists of the following steps:

1. **Pre-annotation**: Prior to the formal annotation, multiple rounds of pre-annotation were conducted to refine the annotation standards. Each round of pre-annotation involved 100 samples. Six annotators were trained and provided with the evaluation criteria and examples. After becoming familiar with the criteria, they rated the provided samples. After each round, the inter-annotator agreement was calculated. Based on the inconsistent parts, the relevant annotation rules were revised according to the discussion results. This process continued for a total of 5 rounds. Eventually, four annotators with the highest consistency ratings were selected to move onto the formal annotation stage.

2. **Formal annotation**: The formal annotation process also used a multi-round iterative mode, with each Q&A sample annotated by two annotators, A and B. Any uncertainties or disagreements between annotators were recorded in Tencent documents. A and B initially address their disagreement, and the project leader quantified the consistency of the annotation results, identifying and documenting the areas where inconsistencies and uncertainties were found. These issues were then discussed with experts to reach a consensus. After the discussion, A was responsible for modifying the annotation to form the final version of the annotation file.

3. **Data verification and integration**: After completing the formal annotation, the project leader checked whether the annotated data conforms to the latest version of the annotation criteria, as well as whether there were missing scores and other issues. Once confirmed, all annotated data was integrated, and key information about the data set, such as sample size and score distribution in each dimension, is summarized.

## C EXAMPLES OF MISLABELED DATA POINTS AND QUALITY CONTROL

Sample data points that were mislabelled are presented in Table 6. Our approach of quality control involved training the FinBERT model on the complete dataset for a limited duration of 5 epochs,

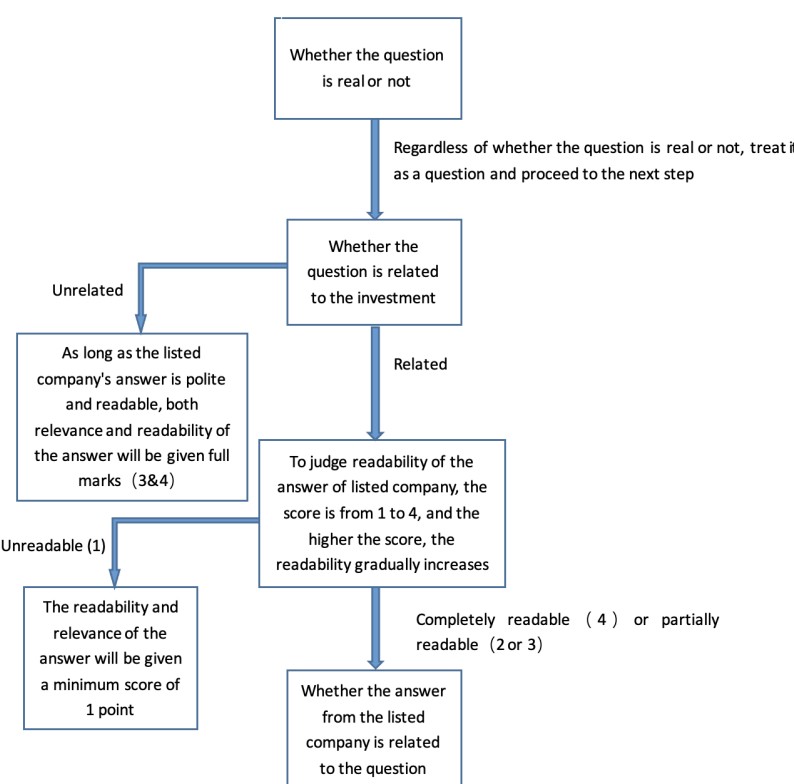

Figure 5: Guideline of the annotation process.

Table 6: Sample data points that were mislabeled.

| Question | Answer | Label |
|---|---|---|
| 您好，请问大股东股权质押的平仓线是多少？谢谢

Hello, may I ask what is the liquidation line for the major shareholder's share pledge? Thank you. | 尊敬的投资者，公司未收到大股东出现平仓情况的通知。感谢您对贵人鸟的关注。

Dear investors, the company has not received any notification regarding the occurrence of forced liquidation of major shareholders. Thank you for your attention to Guirenniao. | 1 |
| 公司领导为什么不学李东生，在大会上说公司估值太低？常年二三元的股价，有损公司形象？
Why don't company leaders learn from Dongsheng Li and say at the conference that the company is undervalued? The stock price has been around two to three yuan for years, damaging the company's image? | 您好，感谢您对公司的关注。

Hello, thank you for your interest in the company. | 3 |
| 董秘你好,云海金属经过多年发展已经成长为一个集科研,垂直产业一体化的公司,不仅仅是一个单一有色金属的公司,建议公司更名,一是拓展自身的发展愿景,二是让更多的投资者知道了解云海,融资助力云海更上一层楼。建议更名"云海科技",或者"云海创新",改名绝不是蹭热度,混时尚,而是极具战略意义的宣言。希望公司采纳,谢谢。
Dear Secretary, Yunhai Metals has grown into a company that integrates scientific research and vertical industry integration after years of development. It is no longer just a company that deals with non-ferrous metals. It is suggested that the company change its name to expand its own development vision and let more investors know about Yunhai, aiding in financing and helping Yunhai reach a new level. The suggested names are "Yunhai Technology" or "Yunhai Innovation". Changing the name is not about chasing trends or blending into the fashion scene but is a declaration of significant strategic meaning. I hope the company will consider this suggestion. Thank you. | 好的，谢谢您的建议。

Okay, thank you for your suggestion. | 1 |

Table 7: Links for different PLMs used in our experiments.

| Model name | Link |
|---|---|
| BERT | https://huggingface.co/google-bert/bert-base-chinese |
| RoBERTa | https://huggingface.co/hfl/chinese-roberta-wwm-ext |
| FinBERT | https://github.com/valuesimplex/FinBERT |
| Mengzi-fin | https://huggingface.co/Langboat/mengzi-bert-base-fin |
| DKPLM | https://huggingface.co/alibaba-pai/ pai-dkplm-financial-base-zh |
| SBERT-nli | https://huggingface.co/uer/sbert-base-chinese-nli |
| RoBERTa-extractive-qa | https://huggingface.co/uer/roberta-base-chinese-extractive-qa |
| BERT (Large) | https://huggingface.co/yechen/bert-large-chinese |
| RoBERTa (Large) | https://huggingface.co/hfl/chinese-roberta-wwm-ext-large |

Table 8: Hyperparameters for BERT-based models. The embedding size is 768 for the base model and 1024 for the large model. Other hyperparameters are shared across both configurations.

| Parameter | Value | Parameter | Value |
|---|---|---|---|
| learning rate | 2e-5 | embedding size (base) | 768 |
| dropout rate | 0.1 | embedding size (Large) | 1024 |
| number of epochs | 30 | batch size | 32 |
| lr decay | 0.01 | max length | 256 |

a precautionary measure to prevent overfitting. Subsequently, this trained model was utilized to predict the entire dataset. The data points where the model's predictions were inconsistent with the labels annotated by experts were extracted and sorted in ascending order according to their predicted probabilities. These sorted data points were then presented to experts for further evaluation. This method aligns with the CL framework, which emphasizes the identification and correction of label errors in a structured manner (Northcutt et al., 2021b). By utilizing predicted probabilities as indicators of prediction confidence, our method facilitated a focused review of the most challenging samples for the model. This strategy allowed experts to concentrate on instances where the model faced difficulty in classification. The experts could then revisit the labeling instructions and make necessary corrections. The revised annotations were then utilized to retrain the model, contributing to an refined dataset and subsequently improved model performance.

The process was conducted for the *answer relevance* task over two rounds. In the first round, 201 QA pairs were identified where the model predicted incorrectly. We had the annotators re-evaluate and correct the labels for 91 of them. In the second round, we identified an additional 151 data that required re-evaluation, resulting in 38 corrected labels. The distribution of labels for this task presented in this paper was obtained after the quality control process.

## D PRE-TRAINED LANGUAGE MODEL

Table 7 provides links to specific PLMs we used in this study.

## E HYPERPARAMETER SETTINGS

Table 8 presents the hyperparameter settings of our BERT-based model, which were kept fixed across all experiments except from the embedding size. These hyperparameters were initialized based on prior work in the literature. During our initial experiments, we also fine-tuned these parameters to strike a suitable balance between model complexity and performance using the validation set from the preprocessed dataset. In particular, the embedding size for the BERT-base model was set to 768, while it was set to 1024 for BERT-large model. AdamW (Loshchilov & Hutter, 2018) was used as the optimizer in all experiments as we found that AdamW exhibited better stability and faster convergence compared to stochastic gradient descent. The experiments were conducted using two NVIDIA RTX 3090 GPUs.

# F EXAMPLE PROMPT FOR LLM-BASED EXPERIMENTS

The prompts of LLMs are shown in table 9, which include two versions: "Instructions to Score Answers" and "Extended Prompts Requiring Analysis." The former instructs the model to directly output relevance scores, while the latter requires additional analysis output. The prompts provided to LLMs were designed as a classification task. We instructed the model to assign a score of 1, 2, or 3, with each score corresponding to a specific level of relevance. This design aligns with a typical classification task, where the model is required to categorize the answers based on predefined criteria. The scoring system was explicitly explained to the model, and the model's outputs were treated as categorical classifications, thus ensuring that the evaluation remained consistent with the other tasks.

Table 9: Prompt for LLM-based experiments. The text in purple instructs the model to provide an analysis of its decision-making process alongside the relevance scores. This instruction was only included in the prompts for the second experiment to gather insights into LLMs' rationale for scoring answer relevance.

---

**INSTRUCTIONS TO SCORE ANSWERS**

## Instructions

假设您是一名经验丰富的金融信息分析师，您需要从信息披露的角度来判断Q&A数据的质量。请您根据以下评分标准，对每个问题的回答进行判断并给予相应的分数，并按照'回答相关性评分为：...'的格式输出评分：

(Translation: Assume you are an experienced financial information analyst. You need to evaluate the quality of Q&A data from the perspective of information disclosure. Please judge each answer based on the following scoring criteria, assign a corresponding score, and output the score in the format: "The relevance score of the answer is: ...".)

## Evaluation Criteria

评分标准：

1分：回答与提问完全不相关（完全不相关指所问问题一个也没有回答）

2分：部分相关（部分相关指回答了部分问题）或难以判断

3分：回答与提问完全相关（完全相关指回答了提问中的全部问题）

说明：回答相关性要求公司对投资者的提问一一进行正面回复。所谓相关，指回答正面回答了提问者的问题，不回答或者是回答与问题主题有关但并未正面回答提问都为无关。正面回复了提问者的所有问题，则为完全相关，得3分。所有的问题都没有正面回复，则为完全不相关，得1分。正面回答了提问中的部分问题，则为部分相关，部分相关或是难以判断的回答得2分。如果一个回答能回答多个问题，也为全部相关，得3分；如果提问中提到的问题与公司无关，公司只需要否定，不需要回答后续问题。

(Translation: Scoring criteria:

1 point: The answer is completely unrelated to the question (completely unrelated means none of the questions were answered).

2 points: Partially related (partially related means some questions were answered) or difficult to judge.

3 points: The answer is fully related to the question (fully related means all questions were answered).)

## Input

问题(Question): {question}

回答(Answer): {answer}

---

**EXTENDED PROMPTS REQUIRING ANALYSIS**

## Instructions

假设您是一名经验丰富的金融信息分析师，您需要从信息披露的角度来判断Q&A数据的质量。请您根据以下评分标准，对每个问题的回答进行判断并给予相应的分数，并按照'分析过程为：...因此，回答相关性评分为：...'的格式输出评分：

(Translation: Assume you are an experienced financial information analyst. You need to evaluate the quality of Q&A data from the perspective of information disclosure. Please judge each answer based on the following scoring criteria, assign a corresponding score, and output the score in the format: "Analysis process: ..., therefore, the relevance score of the answer is: ...".)

## Evaluation Criteria
评分标准：
1分：回答与提问完全不相关（完全不相关指所问问题一个也没有回答）
2分：部分相关（部分相关指回答了部分问题）或难以判断
3分：回答与提问完全相关（完全相关指回答了提问中的全部问题）
说明：回答相关性要求公司对投资者的提问一一进行正面回复。所谓相关，指回答正面回答了提问者的问题，不回答或者是回答与问题主题有关但并未正面回答提问都为无关。正面回复了提问者的所有问题，则为完全相关，得3分。所有的问题都没有正面回复，则为完全不相关，得1分。正面回答了提问中的部分问题，则为部分相关，部分相关或是难以判断的回答得2分。如果一个回答能回答多个问题，也为全部相关，得3分；如果提问中提到的问题与公司无关，公司只需要否定，不需要回答后续问题。
(Translation: Scoring criteria:
1 point: The answer is completely unrelated to the question (completely unrelated means none of the questions were answered).
2 points: Partially related (partially related means some questions were answered) or difficult to judge.
3 points: The answer is fully related to the question (fully related means all questions were answered).)

## Input
问题(Question): {question}
回答(Answer): {answer}

# G  DATA DIVERSITY ASSESSMENT

The questions of our dataset cover a wide range of topics, including project and investment inquiries (e.g., questions about specific project, investment impacts and progress), financial and stock performance (e.g., inquiries about stock price movements, shareholding structures and potential market manipulation), operational and production issues (e.g., questions related to production plans and operational status during the pandemic), market and competitive analysis (e.g., concerns about market share and competition in specific sectors), raw material and supply chain (e.g. inquiries about material usage and supply chain issues), etc. The questions also vary in depth, ranging from surface-level queries about basic facts (e.g., number of shareholders) to more detailed questions requiring in-depth responses, such as the implications of specific projects on future earnings or strategic decisions regarding market expansion. These examples indicate that our dataset is not limited to a single type of question but instead captures a broad spectrum of investor concerns, ranging from financial and stock performance to operational details and market competition.

**Case 1: About project and investment inquiries**
问题（Question）：董秘您好，请问大连项目是否已经投产，会对今年业绩产生积极影响吗？（Dear Secretary of the Board, could you please confirm if the Dalian project has already commenced production? Will it have a positive impact on this year's performance?）
答案（Answer）：尊敬的投资者，目前大连项目处于试运行阶段，试运行阶段会产生部分的收益，预计本年度不会对公司业绩产生较大影响。谢谢！（Dear Investors, the Dalian project is currently in the trial operation phase. This phase will generate some revenue, but it is not expected to have a significant impact on the company's performance this year. Thank you!）

**Case 2: About financial and stock performance**
问题（Question）：火炬是否经营正常，股价最近很低迷，是否存在恶意卖空，陶瓷材料新业务预计什么时候可以产生投资收益，谢谢！（Is the Torch project operating normally? The stock price has been sluggish recently—could there be malicious short selling? When is the new ceramics materials business expected to generate investment returns? Thank you!）

答案（Answer）：您好！公司经营状况一切正常。另外定增项目已完成三条（总共六条）材料生产线的安装调试工作，已进入试生产阶段，待相关鉴定及验证完成后，可释放5吨/年的产能，形成新的利润增长点。谢谢关注。（Hello! The company's operations are running smoothly. Additionally, three out of six production lines for the materials in the private placement project have completed installation and commissioning and have entered the trial production stage. Once the relevant assessments and verifications are completed, a production capacity of 5 tons per year will be released, creating a new profit growth point. Thank you for your attention.）

**Case 3: About market and competitive analysis**

问题（Question）：你好，近期菜油和玉米期货大涨，能否介绍下公司油脂业务方面的情况和市场占有率，以及玉米大宗货物交易方面的业务情况和中国玉米网的市场地位。（Hello recently, there has been a significant increase in the futures prices of rapeseed oil and corn. Could you please provide an overview of the company's operations in the oils and fats business, including its market share? Additionally, could you explain the company's business activities in bulk corn trading and the market position of China Corn Network?）

答案（Answer）：尊敬的投资者您好。公司油脂业务经营主体为厦门东方银祥油脂有限公司和上海东瑞贸易有限公司，主营产品为豆油（一级豆油，四级豆油、大豆原油）、菜油（一级菜油，四级菜油，拍卖油）、豆粕、菜籽粕、磷脂油等，产品销售覆盖四川、重庆、贵州、广西、安徽等省市。公司玉米收购销售业务经营模式主要包括自营业务、港口中转业务和政策性粮贸易业务。公司旗下中国玉米网（中国玉米供应链服务平台）是国内玉米领域最大的信息集成平台，平台积极促进贸易对接，使玉米产业链上下游企业在线上更便捷地完成交易，并提供全面高效的线上数据信息服务。该平台自2020年2月上线至2020年6月末累计成交总额为15.12亿元。谢谢。（Dear Investor, thank you for your inquiry. The company's oil business is primarily managed by Xiamen Dongfang Yixiang Oil Co., Ltd. and Shanghai Dongrui Trading Co., Ltd. The main products include soybean oil (first-grade soybean oil, fourth-grade soybean oil, crude soybean oil), rapeseed oil (first-grade rapeseed oil, fourth-grade rapeseed oil, auction oil), soybean meal, rapeseed meal, lecithin oil, and others. These products are sold in various provinces and cities, including Sichuan, Chongqing, Guizhou, Guangxi, and Anhui. The company's corn procurement and sales business operates through various models, including self-operated business, port transshipment business, and policy-based grain trade business. The China Corn Network (China Corn Supply Chain Service Platform), a subsidiary of the company, is the largest information integration platform in the domestic corn industry. The platform actively promotes trade matching, enabling upstream and downstream enterprises in the corn industry chain to conduct transactions more conveniently online while providing comprehensive and efficient online data and information services. From its launch in February 2020 to the end of June 2020, the platform achieved a total transaction value of 1.512 billion yuan. Thank you.）

The variety ensures that the dataset is robust and capable of supporting comprehensive analysis across multiple dimensions of corporate communication and financial disclosure quality. This diversity enhances the value of our dataset, making it a valuable resource for understanding the different facets of company-investor interactions.

