# OpenReview forum: "FinTruthQA: A Benchmark Dataset for Evaluating the Quality of Financial Information Disclosure"
_ICLR.cc/2025/Conference — Submitted to ICLR 2025_

### Official Review · Reviewer_fDei · 2024-10-29

**Soundness:** 3
**Presentation:** 3
**Contribution:** 2
**Rating:** 6
**Confidence:** 3

**Summary:**

The paper introduces FinTruthQA, a new benchmark dataset developed to evaluate the quality of financial information disclosures on interactive platforms for investors in Chinese stock exchanges (Shanghai Stock Exchange and Shenzhen Stock Exchange). The dataset comprises 6,000 real-world Q&A entries and assesses disclosure quality across four key criteria: question identification, question relevance, answer readability, and answer relevance. The authors benchmark various machine learning (ML) and pre-trained language models (PLMs) and examine their effectiveness on these tasks.

**Strengths:**

The authors address the crucial challenge of evaluating the quality of financial disclosures on platforms where companies communicate with investors. The interactive nature of these disclosures and their unstructured format make this a valuable problem.

By focusing on investor-platform Q&A interactions, the paper explores a unique and under-researched form of financial communication that differs from more structured formats like reports or calls.

The experiments cover a wide range of ML and PLM-based models, and the paper offers insights into their performance, especially emphasizing the limitations of current models in answer readability and relevance tasks. This provides a strong foundation for future improvements.

**Weaknesses:**

Although the dataset focuses on investor Q&A in the Chinese stock exchange context, the lack of testing across different regulatory and cultural contexts limits its generalizability. Without broader applicability, the impact of FinTruthQA as a global benchmark remains restricted.

The reliance on human annotation for subjective criteria (e.g., answer readability and relevance) introduces potential biases. While the authors conducted quality control, it is unclear how consistently these subjective labels would hold across diverse annotator groups or application settings.

Although the paper provides an overview of performance results, it lacks an in-depth analysis of errors made by models. Understanding specific failure cases would offer valuable insights into the dataset’s complexity and guide future model improvements.

Many releted papers are not refered, such as
The FinBen: An Holistic Financial Benchmark for Large Language Models
FinTral: A Family of GPT-4 Level Multimodal Financial Large Language Models
Large Language Models as Financial Data Annotators: A Study on Effectiveness and Efficiency
FinCon: A Synthesized LLM Multi-Agent System with Conceptual Verbal Reinforcement for Enhanced Financial Decision Making
Open-FinLLMs: Open Multimodal Large Language Models for Financial Applications

**Questions:**

See above

---

> ### Author Response · Authors · 2024-11-24
>
> We appreciate your time and effort in reviewing our work and these insightful comments,  which have helped us improve our manuscript. We provide a list of responses below:
> (W=Weaknesses; RW=Response to Questions)
>
> ---
>
> **W1**: Although the dataset focuses on investor Q&A in the Chinese stock exchange context, the lack of testing across different regulatory and cultural contexts limits its generalizability. Without broader applicability, the impact of FinTruthQA as a global benchmark remains restricted.
>
> **RW1**: We appreciate your insightful feedback and the opportunity to address this concern regarding the generalizability of FinTruthQA.
> We would like to highlight that the proposed FinTruthQA dataset is rooted in two novel platforms within China’s financial disclosure system, specifically the investor Q&A platforms provided by the Shanghai Stock Exchange (SSE) and Shenzhen Stock Exchange (SZSE). These platforms represent a unique and innovative regulatory approach, enabling direct communication between investors and listed companies. To our knowledge, **similar platforms are not widely observed in other regulatory systems globally**, making FinTruthQA a pioneering effort to study and analyze regulated corporate-investor interactions in this context.
>
> While the platform itself is unique to the Chinese financial system, the methodologies and principles underpinning FinTruthQA are not confined to any single language or regulatory framework. The annotation framework, criteria, and pipeline we developed are inherently **language-agnostic** and **adaptable**. All the code and processes were designed for adaptability, enabling their easy application across various languages and cultural environments. The detailed description of the criteria, coupled with our inclusion of advanced machine learning and pre-trained models, provides a replicable framework for extending FinTruthQA to other regions and contexts. For example, the criteria employed (question identification, question relevance, answer readability, and answer relevance) are universally applicable to similar platforms worldwide. By demonstrating the framework’s efficacy in one regulatory context, we establish a proof of concept that can inspire similar efforts in other markets, thereby expanding its generalizability over time.
>
>
> ---
>
> **W2**: The reliance on human annotation for subjective criteria (e.g., answer readability and relevance) introduces potential biases. While the authors conducted quality control, it is unclear how consistently these subjective labels would hold across diverse annotator groups or application settings.
>
> **RW2**: We acknowledge your concerns regarding the potential subjectivity in our manual annotation process. To address this, we have established a rigorous annotation procedure, as detailed in Section B of our appendix. All annotators underwent multiple rounds of training and pre-annotation exercises, comprising five iterative sessions. During these, feedback was provided to refine annotators’ understanding and agreement levels on subjective criteria. To ensure high-quality annotation, only the top four (out of six) annotators, based on their **inter-annotator agreement (IAA) scores** from these rounds, were selected for the formal annotation phase. After several rounds of quality control, the IAA rate reached **85.52%**, demonstrating a strong level of agreement among the annotators and the robustness and consistency of our annotation process. Comprehensive annotation guidelines were developed and included in the supplementary material.
>
> ---

---

> > ### Author Response · Authors · 2024-11-24
> >
> > **W3**: Although the paper provides an overview of performance results, it lacks an in-depth analysis of errors made by models. Understanding specific failure cases would offer valuable insights into the dataset’s complexity and guide future model improvements.
> >
> > **RW3**: We appreciate your suggestion to include a more detailed error analysis and would like to clarify that the paper already includes a preliminary error analysis (Table 5 of the original manuscript). However, to address your feedback comprehensively, we expanded and refined this analysis in the revised manuscript.
> >
> > According to Table 5 in our manuscript, GPT-4 often fails in relevance tasks due to its tendency to match keywords without understanding the deeper intent of the question (e.g., identifying partial keyword overlaps as highly relevant responses). Also, it fails to associate interchangeable concepts in certain contexts (e.g. "employee benefits" and "salaries). Another notable challenge was its inability to appropriately handle answers marked as “unclear”, misinterpret "unclear" responses as irrelevant rather than acknowledging them as valid answers under uncertainty. This misinterpretation reduces its effectiveness in contexts where acknowledging ambiguity is crucial, particularly in financial domains where definitive answers are not always possible. Addressing these gaps would require an enhanced ability to process uncertainty as a legitimate feature of certain types of responses.
> >
> > ---
> >
> > **W4**: Many releted papers are not refered, such as The FinBen: An Holistic Financial Benchmark for Large Language Models FinTral: A Family of GPT-4 Level Multimodal Financial Large Language Models Large Language Models as Financial Data Annotators: A Study on Effectiveness and Efficiency FinCon: A Synthesized LLM Multi-Agent System with Conceptual Verbal Reinforcement for Enhanced Financial Decision Making Open-FinLLMs: Open Multimodal Large Language Models for Financial Applications
> >
> > **RW4**: Thank you for your valuable feedback highlighting the omission of related works in our manuscript. To address this, we have added a detailed section in our manuscript to thoroughly discuss these critical works and their connections to our study:
> >
> > *Several studies have demonstrated the potential of large language models (LLMs) in financial contexts. FinBen[1] introduced a comprehensive benchmark to evaluate the performance of LLMs across diverse financial tasks, including information extraction, textual analysis, and decision-making. Similarly, Open-FinLLMs[5] presented a suite of multimodal models designed to address the limitations of existing financial LLMs in handling tabular and time-series data. On the other hand, FinTral[2] introduces a multimodal framework built on Mistral-7B, integrating text, tables, and images for comprehensive financial analysis. The referenced work[3] explored the use of LLMs as financial data annotators, comparing their performance against human experts and crowdworkers for tasks such as relation extraction in financial documents. FinCon[4] introduces an innovative multi-agent framework tailored for financial decision-making tasks, such as single-stock trading and portfolio management.*
> >
> >
> > ---
> > **References:**
> >
> > [1] Xie Q, Han W, Chen Z, et al. The finben: An holistic financial benchmark for large language models[J]. arXiv preprint arXiv:2402.12659, 2024.
> >
> > [2] Bhatia G, Nagoudi E M B, Cavusoglu H, et al. Fintral: A family of gpt-4 level multimodal financial large language models[J]. arXiv preprint arXiv:2402.10986, 2024.
> >
> > [3] Aguda T, Siddagangappa S, Kochkina E, et al. Large Language Models as Financial Data Annotators: A Study on Effectiveness and Efficiency[J]. arXiv preprint arXiv:2403.18152, 2024.
> >
> > [4] Yu Y, Yao Z, Li H, et al. FinCon: A Synthesized LLM Multi-Agent System with Conceptual Verbal Reinforcement for Enhanced Financial Decision Making[J]. arXiv preprint arXiv:2407.06567, 2024.
> >
> > [5] Xie Q, Li D, Xiao M, et al. Open-finllms: Open multimodal large language models for financial applications[J]. arXiv preprint arXiv:2408.11878, 2024.
> >
> > ---

---

> > ### Comment · Reviewer_fDei · 2024-11-25
> > **reviewer**
> >
> > I have read the comments and raise my score.

---

### Official Review · Reviewer_u6Wu · 2024-11-03

**Soundness:** 3
**Presentation:** 3
**Contribution:** 2
**Rating:** 5
**Confidence:** 4

**Summary:**

The paper introduces a dataset designed to assess the quality of financial information disclosure using natural language processing (NLP) techniques. This dataset, FinTruthQA, consists of 6,000 annotated Q&A entries from Chinese stock exchanges, focusing on four evaluation criteria: question identification, question relevance, answer readability, and answer relevance. The authors benchmarked various language models, finding that while models perform well in identifying and categorizing questions, they struggle with answer readability and relevance.

**Strengths:**

1. Dataset: The paper introduces a  dataset, FinTruthQA, specifically tailored for assessing financial information disclosure quality, covering critical criteria such as question relevance and answer readability.

2. Real-world Applicability: The dataset and methods are designed with practical applications in mind, aiding auditors, analysts, and regulators in real-time monitoring and enhancing transparency in financial disclosures.

**Weaknesses:**

1. Title Discrepancy: The title of the paper suggests a focus on the truthfulness of the responses within the dataset. However, the actual content primarily addresses other aspects such as readability and relevance. A title that more accurately reflects the paper's content could improve its alignment with the reader's expectations.

2. Baseline Methodologies for Readability: The paper could be strengthened by including more established baselines for evaluating readability, such as the Gunning fog index. For a comprehensive view on this, the authors might consider reviewing literature from targeted workshops like the TSAR workshop on readability (https://tsar-workshop.github.io/).

3. Label Distribution and Utility: The current label selection, particularly for answer readability, seems suboptimal. Labels 2 and 3 are underrepresented, appearing in less than 3% of the QA pairs. This distribution is problematic, as evidenced by the classification challenges shown in Figure 4, where Label 4 is frequently misclassified as Label 1, but rarely as Labels 2 or 3. Refining these categories could enhance the model's learning and classification accuracy.

4. Definition of Readability: The paper's definition of readability focuses on ease of understanding and quality of writing. However, readability could be more effectively assessed by considering additional components such as structure, clarity, style, and grammar. A more detailed framework for readability would likely improve the assessment’s depth and utility.

**Questions:**

None

---

> ### Author Response · Authors · 2024-11-24
>
> We appreciate your time and effort in reviewing our work and these insightful comments,  which have helped us improve our manuscript. We provide a list of responses below:
> (W=Weaknesses; RW=Response to Questions)
>
> ---
>
> **W1**: Title Discrepancy
>
> **RW1**: Thank you for your thoughtful feedback regarding the title discrepancy in our paper. As outlined in the introduction of our paper, the Q&A interactions analyzed are sourced from platforms regulated by the China Securities Regulatory Commission, where companies are legally obligated to ensure the accuracy of their disclosures. Within this highly regulated framework, we define "truthfulness" not merely as factual accuracy but as a broader concept encompassing factual compliance, informational completeness, and relevance to investors' queries. Companies, while meeting their legal obligations, often provide responses that are indirect or minimally informative. This regulatory environment affects our method of analyzing responses, concentrating on aspects like readability and relevance to assess the quality of disclosure.
>
> Recognizing that the original introduction may not have sufficiently linked the context to the dimensions of readability and relevance, we have revised the introduction to clarify this connection as follows:
> *The regulatory framework governing these platforms ensures that listed firms are legally obligated to provide truthful and accurate disclosures. However, companies often respond with limited or indirect information, aiming to satisfy regulatory requirements without fully addressing investor concerns. This situation presents a distinct challenge in assessing the quality of disclosures, which extends beyond simple factual accuracy to include dimensions such as relevance, readability, and informational completeness. These aspects collectively shape the "truthfulness" of disclosures in this context.*
>
> In response to the reviewer’s suggestion to better align the title with the paper’s content, we propose to revise the title to: **"FinDisQA: A Benchmark Dataset for Evaluating the Quality of Financial Information Disclosure**". This title more accurately reflects the focus of our work on assessing the quality of disclosure using criteria such as readability and relevance, without narrowly emphasizing "truthfulness." We will implement this title revision upon acceptance of the paper.
>
> ---
>
> **W2**: Baseline Methodologies for Readability
>
> **RW2**: Thank you for your suggestion to include more established readability baselines, such as the Gunning Fog Index, in our evaluation. We acknowledge the value of traditional readability metrics in many contexts; however, our study, FinTruthQA, specifically focuses on evaluating financial Q&A data, where domain-specific nuances and contextual readability play a crucial role.
>
> Traditional indices like the Gunning Fog Index rely heavily on surface-level linguistic features (e.g., sentence length, syllable count) which do not adequately capture the complexities of financial discourse. Studies[1] have shown that surface-level readability scores often do not correlate well with human-judged readability, particularly for domain-specific texts. FinRAD[2] also highlighted the limitations of traditional readability indices when applied to financial texts. For instance, financial answers often include numerical data, technical terms, and abbreviations that are inherently challenging to parse but essential for conveying accurate information. These aspects are not well-addressed by traditional readability metrics.
>
> As emphasized in our paper, we adopted a fine-grained, domain-specific approach by annotating answer readability using human-labeled scores that account for clarity, coherence, and contextual appropriateness in financial communication (Sections 3.3 and 4.3.3). This methodology aligns more closely with recent trends observed in workshops such as TSAR, where modern, context-aware readability measures are favored over traditional indices. For example, the TSAR workshop proceedings highlight the use of neural and task-specific methods for assessing text simplification and readability rather than relying solely on formula-based measures. FinRAD[2] also used neural-based models to assess the readability of financial definitions.
>
> Furthermore, our experiments demonstrate that advanced NLP techniques, such as FinBERT and RoBERTa-Extractive-QA, outperform traditional methods in readability evaluation tasks (Table 3). These models leverage pre-trained embeddings to effectively account for the semantic and structural intricacies of financial texts, providing a more reliable and nuanced assessment than formulaic indices could offer.

---

> ### Author Response · Authors · 2024-11-24
>
> That said, we understand the potential value of including traditional metrics for comparative purposes. We are willing to incorporate the Gunning Fog Index and similar readability baselines in a supplementary analysis for the final version of the paper, allowing for a broader comparison across methodologies.
>
> ---
>
> **W3**: Label Distribution and Utility
>
> **RW3**: We appreciate your insightful observations on the label distribution and its implications for model performance. Although Labels 2 and 3 appear less frequently, they provide critical insights into intermediate states of readability. This skew reflects the real-world data distribution in financial disclosures, where companies often aim to provide answers that are either very concise (Label 1) or very clear (Label 4). Removing or consolidating these labels would risk losing granularity in the quality assessment, which is especially important for developing nuanced NLP models. While this imbalance introduces challenges for classification models, it is representative of the actual data characteristics.
>
> ---
>
> **W4**: Definition of Readability
>
> **RW4**: We appreciate your detailed observation regarding the definition of readability in our paper. In our current framework, readability is evaluated based on ease of understanding, i.e. whether the content is clear, concise, and free of ambiguities. This choice was guided by the practical constraints and the intended applicability of the FinTruthQA benchmark in assessing financial Q&A disclosures. These dimensions align with the immediate goals of ensuring transparency and accessibility for retail investors. However, we acknowledge that readability encompasses broader attributes such as structure, clarity, style, and grammar. Our evaluation criterion implicitly considers these aspects through the scoring system (e.g., levels indicating the presence of ambiguities). We have enhanced our explanation in the revised manuscript to explicitly convey this comprehensive interpretation:
>
> *Answer readability: This criterion evaluates how easily an answer can be understood and whether it is well-written.  It considers multiple dimensions, including structural coherence, clarity of information, stylistic appropriateness, and grammatical correctness, ensuring a comprehensive evaluation of the answer's quality.*
>
> ---
> **References:**
>
> [1]Pitler E, Nenkova A. Revisiting readability: A unified framework for predicting text quality[C]//Proceedings of the 2008 conference on empirical methods in natural language processing. 2008: 186-195.
>
> [2]Ghosh S, Sengupta S, Naskar S K, et al. FinRAD: Financial Readability Assessment Dataset-13,000+ Definitions of Financial Terms for Measuring Readability[C]//Proceedings of the 4th Financial Narrative Processing Workshop@ LREC2022. 2022: 1-9.
>
> ---

---

> ### Author Response · Authors · 2024-11-26
> **Sincerely Looking Forward to More Discussions**
>
> Dear Reviewer u6Wu,
>
> Thank you for your valuable feedback. We have carefully addressed all your concerns and deeply appreciate your thoughtful suggestion to revise the manuscript.
>
> We clarified the title alignment, opting for a revised title emphasizing disclosure quality rather than "truthfulness." For readability baselines, we prioritized domain-specific, context-aware metrics over traditional formulas, offering to include traditional metrics for comparative analysis in supplementary materials. We addressed concerns about label distribution, highlighting its reflection of real-world data while retaining granularity for nuanced NLP modeling. Additionally, we expanded the definition of readability, ensuring a comprehensive evaluation framework.
>
> We hope these revisions comprehensively address your comments and are happy to discuss any additional concerns you may have.
>
> Best regards.

---

> ### Comment · Reviewer_u6Wu · 2024-11-26
>
> I thank the authors for their response. I have increased my score for "Soundness" score based on their response.
>
> I still have some reservations about the **Label Distribution and Utility**, as labels are ordinal and misclassification is not just 1 level above or below the original label. If other reviewers suggest acceptance, I would not object to their decision.

---

> > ### Author Response · Authors · 2024-11-27
> > **Response to Reviewer Comments on Label Distribution and Utility**
> >
> > We thank the reviewer for the follow-up comment and for raising the soundness score. The definition of the labels and their granularity was carefully crafted by a senior accounting professor (one of our coauthors) with over 20 years of experience in financial information disclosure research, following multiple rounds of in-depth discussions. The label design prioritizes ease of use and explainability within the disclosure domain from an accounting perspective. Consequently, we have decided to retain the current annotation setting to better align with the specific needs of the accounting domain. The imbalanced distribution accurately reflects the real-world distribution of our data, while the intermediate labels capture nuanced yet critical behaviors associated with financial information disclosure.
> >
> > We are happy to address any remaining concerns and would greatly appreciate it if the reviewer could reevaluate the overall score in light of this context. Thank you very much!

---

> > > ### Comment · Reviewer_u6Wu · 2024-11-27
> > >
> > > I have tried my best to recreate the confusion metric I was referring to here in the markdown for the convenience of discussion.
> > >
> > > | True Label \ Predicted Label | 1   | 2   | 3   | 4   |
> > > |------------------------------|-----|-----|-----|-----|
> > > | 1                            | 133 | 1   | 0   | 64  |
> > > | 2                            | 1   | 0   | 0   | 5   |
> > > | 3                            | 4   | 1   | 0   | 15  |
> > > | 4                            | 20  | 0   | 0   | 673 |
> > >
> > > ---
> > >
> > > Thank you for your reply. I respect your thought process around creating these 4 labels and appreciate the domain-centric approach you've taken. However, I wanted to clarify my thoughts that might help refine the interpretation and design of your work/model.
> > >
> > > Looking at the confusion matrix, it seems that while 4 labels are theoretically (from a domain perspective) meaningful, the distribution and behavior in the results raise some concerns about their utility from a modeling perspective:
> > >
> > >  - **Distribution of Labels**: The sub-matrix for labels 2 and 3 ([\[0, 0], \[1, 0\]]) highlights a significant imbalance or lack of representation, making it difficult for the model to learn and accurately predict these labels. Additionally, column 3 is entirely 0s, and the diagonal values for both labels 2 and 3 are also 0. This suggests that the model struggles with these classes or that there might not be sufficient data for the model to meaningfully differentiate between these labels.
> > >
> > >  - **Misclassification Patterns**: I noticed that labels 1 and 4 frequently get misclassified as 4 and 1, respectively. This indicates that the model is more adept at distinguishing between these two dominant classes but less so for the intermediate classes (2 and 3). From a data perspective, this skewed distribution makes it challenging for the model to take advantage of the full 4-class structure.
> > >
> > >  - **Model Perspective**: From a purely model-centric view (rather than a domain-specific one), the current use of 4 labels appears to limit the model’s power to generalize effectively due to the under-representation of certain classes and their low predictive reliability. A potential improvement could involve approaches like **bootstrapping** or resampling, which could help balance the class distributions and improve the model's ability to leverage these intermediate classes.
> > >
> > > While I understand the rationale for 4 labels might stem from domain-specific considerations, I would encourage exploring ways to address these distributional issues to enhance the model's effectiveness. Alternatively, it may be worth considering whether collapsing or restructuring the labels could provide a more balanced and interpretable classification framework.
> > >
> > > I hope this feedback is helpful and contributes to further improving your already good work.

---

> ### Author Response · Authors · 2024-11-27
> **dataset value and limitation section has been updated**
>
> We sincerely thank the reviewer for their time, effort, and detailed, insightful comments. We acknowledge that the significant imbalance in the data distribution stems from real-world scenarios. In response to the reviewer's concern, we have expanded our discussion in the limitations section to include the following:
>
> ***"Lastly, the question relevance and answer readability tasks suffer from a highly imbalanced dataset, which may limit the model’s ability to effectively learn from and evaluate rare instances. However, we chose to retain this task in the benchmark to reflect the real-world data distribution in financial information disclosure."***
>
> We believe that the decision to collapse the labels or retain the fine-grained labels should be left to the users, depending on their specific applications. While techniques such as sampling or boosting could help mitigate these issues and improve model performance, it is important to emphasize that model optimization is not the primary focus of our work.
>
> **The main contribution of this paper lies in the development of a novel and domain-specific open dataset with practical applications in the financial information disclosure field. This dataset provides a foundation for users to test advanced AI methods and improve performance. Our study establishes a comprehensive baseline, offering a clear and reliable point of comparison for future research.**
>
> We truly appreciate the reviewer’s responsiveness, which provides valuable suggestions for further exploration. We believe this dataset serves as a valuable resource for advancing timely and automated financial disclosure analysis, a critical area in the accounting research domain. We encourage users to build upon our dataset and adopt advanced methodologies to address the challenges highlighted.

---

### Official Review · Reviewer_PYGZ · 2024-11-04

**Soundness:** 3
**Presentation:** 4
**Contribution:** 2
**Rating:** 6
**Confidence:** 4

**Summary:**

The authors discuss the relationship between the quality of disclosure in financial Q&A data and the ability of investors to make informed decisions. In this paper, the author created a new benchmark dataset (FinTruthQA) containing 6,000 manually annotated financial Q&A pairs from Chinese stock platforms. The author benchmarked various approaches including ML-based models, pre-trained language models, and large language models.

**Strengths:**

1. This paper targets a real-world problem in financial markets about how to improve transparency and quality of financial reporting.

2. The annotation process is comprehensive in that the annotation team takes multiple rounds and quality control, comprehensive evaluation using multiple metrics, and documentation of the data collection and processing steps.

3. This paper evaluated multiple types of models (ML, PLM, LLM), including domain-specific models, and continued pre-training models. The author also showed a detailed error analysis and performance comparisons.

4. The experiment results demonstrated that domain-specific models, FinBERT with continued pre-training, perform better on complex tasks. This insight is useful to the domain.

**Weaknesses:**

1. The FinTruthQA dataset is exclusively from Chinese markets. In addition, the proposed dataset focuses only on the Q&A format, excluding other types of financial disclosure (like, long reports, and analytic reports).

2. In question relevance, the label is imbalanced (99.51% positive), and limited number of samples for some categories. This may affect the validity and generalizability of the benchmark results.

3. Limited overall technical contribution to financial Q&A or general domain Q&A.

4. Experiment detail is not comprehensive, such as no details such as temperature were reported.

**Questions:**

See in Weaknesses.

In addition, the authors still use traditional metrics when evaluating. I'm not sure if this is entirely appropriate for evaluation in the financial sector.

---

> ### Author Response · Authors · 2024-11-24
>
> We appreciate your time and effort in reviewing our work and these insightful comments, which have helped us improve our manuscript. We provide a list of responses below:
>
> (Q=Question from Weaknesses and Details Of Ethics Concerns; RQ=Response to Questions)
>
> ---
>
> **The following questions are from Weaknesses**:
>
> **Q1**: The FinTruthQA dataset is exclusively from Chinese markets. In addition, the proposed dataset focuses only on the Q&A format, excluding other types of financial disclosure (like, long reports, and analytic reports).
>
> **RQ1**: We appreciate your observation regarding the dataset's focus on Q&A data from the Chinese markets. The focus on the Chinese market is intentional and addresses a unique gap in financial research, given the unique and underexplored nature of investor Q&A platforms in this regulatory and cultural context. The emphasis on the Q&A format in FinTruthQA was to address the growing prevalence of investor interactive platforms that utilize this format for real-time communication. Unlike static financial disclosures such as annual reports, Q&A exchanges provide an interactive and dynamic medium, often reflecting investor concerns and corporate responsiveness in near real-time. By focusing on this novel format, FinTruthQA fills a significant gap in existing benchmarks, which primarily evaluate static disclosures like earnings calls or annual reports.
>
> ---
>
> **Q2**: In question relevance, the label is imbalanced (99.51% positive), and limited number of samples for some categories. This may affect the validity and generalizability of the benchmark results.
>
> **RQ2**: Thank you for your insightful feedback regarding the imbalanced label distribution and the limited sample size for some categories. While it is true that the dataset exhibits significant class imbalance, this imbalance reflects the actual conditions on investor interactive platforms, where most investor queries on platforms like SSE and SZSE pertain to relevant financial topics. By preserving this distribution, we ensure that the benchmark realistically evaluates the performance of models in real-world financial information disclosure contexts. Artificially modifying this distribution could lead to models optimized for a theoretical scenario, thereby reducing their practical applicability.
>
> While the negative labels are limited, they cover diverse instances of irrelevant questions, ensuring a representative sample of the minority class. We intentionally preserved them in the dataset to enable models to learn to identify rare but critical instances of irrelevant questions.
>
> ---
>
> **Q3**: Limited overall technical contribution to financial Q&A or general domain Q&A.
>
> **RQ3**: We appreciate the reviewer’s comment and would like to clarify and highlight the unique contributions of our work.
>
> The primary contribution of our work lies in the creation of the first novel and annotated dataset related to financial information disclosure, providing a valuable resource for the financial NLP field and contributing to improving financial information disclosure practices in real-world financial markets. As we emphasized in our paper, accurate and transparent financial information disclosure is critical in the fields of accounting and finance, ensuring market efficiency and fostering investor confidence. As a result, we collected data from various platforms and systematically annotated it to construct the first benchmark for evaluating the quality of financial information disclosure.
>
> To ensure the quality of the dataset, we established detailed annotation guidelines and implemented strict quality control throughout the dataset construction process. Additionally, we introduced multiple models to assess their performance on this benchmark. The consistent performance across different models demonstrates the reliability of our approach, while the lower performance on certain tasks highlights the challenging nature of the benchmark. This indicates that our benchmark can serve a valuable resource to promote future research in the field.
>
> Beyond direct application in financial and regulatory domains, our benchmark serves as a valuable resource for academic research in natural language processing, particularly in areas concerning the automated assessment of text quality. By establishing a benchmark, we provide a foundation for further research into improving the accuracy and applicability of NLP techniques across diverse fields, potentially leading to innovations that could benefit a wide range of industries where transparent communication is vital.
>
> ---

---

> > ### Author Response · Authors · 2024-11-24
> >
> > **Q4**: Experiment detail is not comprehensive, such as no details such as temperature were reported.
> >
> > **RQ4**: Thank you for your constructive feedback on the comprehensiveness of the experimental details. The temperature parameter for our model experiments was consistently set to the default value of 1.0. We acknowledge that temperature settings were not explicitly mentioned in our original submission. In our revised version, we have added the hyperparameter setting for LLM-based models to Appendix E:
> >
> > Parameter|Value|Parameter|Value
> > ---|---|---|---
> > temperature|1.0|max new token|512
> >
> >
> >
> > ---
> >
> > **The following questions are from Questions**:
> >
> > **Q5**: The authors still use traditional metrics when evaluating. I'm not sure if this is entirely appropriate for evaluation in the financial sector.
> >
> > **RQ5**: Thank you for your constructive feedback regarding the appropriateness of the metrics employed in the financial sector. In our study, we utilized widely recognized metrics for ordinal classification tasks, including accuracy, F1-score, and Quadratic Weighted Kappa (QWK), to thoroughly evaluate model performance across four criteria. To ensure the metrics were suitable for domain-specific applications, we consulted with our coauthor, a senior professor in accounting, who confirmed that these metrics are well-aligned with the evaluation needs of the financial and accounting sectors. Based on this guidance, we believe the selected metrics are sufficient and appropriate for applications within the accounting domain.
> >
> > ---

---

> ### Comment · Reviewer_PYGZ · 2024-11-25
> **read rebuttal**
>
> I have read the rebuttal and I will keep my score.

---

### Official Review · Reviewer_EAeh · 2024-11-05

**Soundness:** 2
**Presentation:** 3
**Contribution:** 3
**Rating:** 5
**Confidence:** 4

**Summary:**

This paper presents a dataset designed to assess the quality of responses on Chinese stock exchange investor platforms, specifically the Shanghai and Shenzhen exchanges. The FinTruthQA dataset includes 6,000 Q&A entries, each manually annotated across four evaluation criteria: question identification, question relevance, answer readability, and answer relevance. The authors benchmark various NLP models, including machine learning methods, pre-trained language models, and large language models, demonstrating that while models perform well on question-related tasks, challenges persist in evaluating answer readability and relevance.

**Strengths:**

1. Novel Benchmark Dataset: The paper introduces FinTruthQA, a unique and practical dataset for evaluating the quality of financial information disclosure in Q&A format on investor platforms. This dataset fills a gap in financial NLP, providing a structured way to evaluate both question and answer quality, which is valuable for real-world applications in finance.

2. Comprehensive Model Evaluation: The authors benchmark a range of NLP techniques, from traditional ML-based models to pre-trained language models (PLMs) and LLMs, demonstrating the utility and limitations of each in assessing Q&A quality.

**Weaknesses:**

1. Limited Evaluation Settings: The evaluation framework is somewhat restrictive, as only ML-based and PLM-based models are assessed across most tasks, while LLMs are evaluated solely on the answer relevance task in a zero-shot setting. A comprehensive comparison across all tasks is necessary to fully understand the strengths and limitations of ML-based models, PLMs, and LLMs. Moreover, applying only zero-shot results for LLMs on answer relevance is inconsistent with the supervised fine-tuning used for ML and PLM models, which could skew the comparative assessment.

2. Copyright and Data Usage Concerns: The paper lacks clarity regarding copyright compliance for the data sources. It is unclear if legal permissions have been obtained to use and openly release the Q&A entries from stock exchange platforms for research purposes. Explicit details on copyright considerations are essential to ensure the ethical and legal viability of this dataset for the broader research community.

3. Exclusion of Financial LLMs: While the paper evaluates a range of models, it does not include financial domain-specific LLMs, which are highly relevant for the specialized nature of this task. Incorporating financial LLMs would provide valuable insights into the performance gains from domain-specific pre-training and would create a more representative assessment of available model options.

4. Class Imbalance in Dataset: The dataset suffers from significant class imbalance, such as in tasks like question identification, where the majority of samples are labeled as positive.

**Questions:**

See weakness

**Details Of Ethics Concerns:**

The paper lacks clarity regarding copyright compliance for the data sources. It is unclear if legal permissions have been obtained to use and openly release the Q&A entries from stock exchange platforms for research purposes. Explicit details on copyright considerations are essential to ensure the ethical and legal viability of this dataset for the broader research community.

---

> ### Author Response · Authors · 2024-11-24
>
> We appreciate your time and effort in reviewing our work and these insightful comments, which have helped us improve our manuscript. We provide a list of responses below:
>
> (Q=Question from Weaknesses and Details Of Ethics Concerns; RQ=Response to Questions)
>
> ---
> **The following questions are from Weaknesses**:
>
> **Q1**: Limited Evaluation Settings
>
> **RQ1**: We appreciate your feedback highlighting the limitations in scope. We have conducted evaluations on tasks such as question identification, question relevance, answer readability, and answer relevance. Tasks like question identification and question relevance achieved sufficient performance with traditional models (>96% accuracy, as shown in Table 2 of the manuscript), making detailed comparisons unnecessary for these tasks. Instead, we focused on more challenging tasks, such as answer readability and answer relevance, where performance differences between traditional ML-based models and PLM-based models were more pronounced.
>
> Due to resource restrictions, we evaluated LLMs only in a zero-shot setting, prioritizing challenging tasks like answer relevance to demonstrate their generalization potential. We have already emphasized in Section 4.1 of the manuscript that PLM and ML models were evaluated with supervised fine-tuning, while LLMs were tested in a zero-shot setting. This background is included to ensure readers clearly understand the methodological context of the comparisons. As we have released the dataset fully open-accessed, users can easily conduct additional experiments on our dataset.
>
> To address potential confusion, we have revised the limitation section:
>
> *Given the substantial computational cost of training, we limited LLM evaluations to a zero-shot setting for answer relevance. This decision balances feasibility with the need to showcase LLM capabilities. Future studies will incorporate fine-tuned evaluations of LLMs to provide a more consistent comparative framework.*
>
> ---
>
> **Q2**: Copyright and Data Usage Concerns
>
> **RQ2**: We appreciate your feedback highlighting the importance of ensuring clarity regarding copyright compliance and data usage.
> The Q&A data used in this study was obtained from publicly accessible interactive platforms of the Shanghai Stock Exchange (SSE) and Shenzhen Stock Exchange (SZSE), which facilitate public disclosure of communications as part of regulatory requirements. It solely comprises communications between public companies and investors, containing no private or restricted content. As the topics focus on publicly listed companies and the information is required to be openly disclosed, we identified no restrictions on its usage. Our dataset has been developed exclusively for academic purposes, aiming to enhance the efficiency and effectiveness of financial communication through data-driven evaluations of financial information quality.
>
> ---
>
> **Q3**: Exclusion of Financial LLMs
>
> **RQ3**: We appreciate your insightful feedback regarding the financial domain-specific LLMs. As we discussed in Section 4.3.4 (under the "Suboptimal Performance of LLMs in Chinese Financial Contexts" section), we explored the use of Chinese financial-specific LLMs, including **CFGPT2** and **DISC-FinLLM**, which represent state-of-the-art domain-specific financial LLMs tailored for Chinese texts. We decided not to incorporate financial-specific LLMs due to the suboptimal performance they demonstrated in our preliminary experiments. Despite their domain specialization, these models failed to generate accurate and reasonable predictions. For example, CFGPT2 frequently generated responses containing extensive redundant phrases (the input question was repeated multiple times), failing to provide substantive information or contextual judgment. Thus, we decided to exclude them from our evaluation.
>
> We have emphasized the following statement in the revised manuscript:
>
> *Preliminary experiments with Chinese financial-specific LLMs demonstrated significant challenges in output quality. Outputs often exhibited excessive repetition and encoding errors. For instance, CFGPT2 redundantly repeated phrases such as “公司是否存在应披露而未披露的重大其他风险” ("Does the company have any significant undisclosed risks?") multiple times when the phrase was part of the input question. Similarly, encoding issues resulted in outputs that were partially unreadable, further limiting their practical utility.*
>
> ---

---

> > ### Author Response · Authors · 2024-11-24
> >
> > **Q4**: Class Imbalance in Dataset
> >
> > **RQ4**: We appreciate your insight into the challenge of class imbalance in our dataset. While it is true that the dataset exhibits significant class imbalance, this imbalance reflects the actual conditions on investor interactive platforms, where most interactions on these platforms involve genuine investor questions. By preserving this distribution, we ensure that the benchmark realistically evaluates the performance of models in real-world financial information disclosure contexts. Artificially modifying this distribution could lead to models optimized for a theoretical scenario, thereby reducing their practical applicability.
> >
> > Although infrequent samples are underrepresented, they are crucial for evaluating a model's ability to handle edge cases. Incorporating these samples enables the benchmark to evaluate models' sensitivity to subtle distinctions between relevant and irrelevant questions(or non-questions), a key requirement for robust financial applications.
> >
> > ----
> >
> > **The following questions are from Details Of Ethics Concerns**:
> >
> > **Q5**: The paper lacks clarity regarding copyright compliance for the data sources. It is unclear if legal permissions have been obtained to use and openly release the Q&A entries from stock exchange platforms for research purposes. Explicit details on copyright considerations are essential to ensure the ethical and legal viability of this dataset for the broader research community.
> >
> > **RQ5**: Thank you for your feedback. We have incorporated the relevant discussion under RQ2. To further clarify and reinforce our stance, we provide the following points:
> >
> > 1. **Legitimacy of Data Sources:**
> >    All the data utilized in this study were obtained from public disclosure platforms mandated by law. As outlined in Article 78 of the *Securities Law of the People’s Republic of China*, issuers and other obligated entities must disclose accurate, comprehensive, and timely information, which is made publicly accessible to all individuals and organizations. This legal provision guarantees that the data we used are publicly available and legally accessible.
> >
> > 2. **Purpose of Data Usage:**
> >    This research exclusively uses publicly disclosed data for academic purposes, intending to advance knowledge and provide insights without any commercial implications. The *Administrative Measures for the Disclosure of Information by Listed Companies* further highlight that disclosures are intended to protect investors’ rights and enhance market transparency. Using this data for non-commercial academic purposes fully aligns with these objectives.
> >
> > 3. **Copyright Compliance:**
> >    As the data are publicly available due to legal disclosure requirements, they are inherently open for public use. Additionally, we have diligently followed the principle of proper attribution, ensuring all data sources are explicitly cited in the study. The data have not been commercially distributed or republished in a manner that would violate copyright laws.
> >
> > 4. **Commitment to Research Ethics:**
> >    We acknowledge the critical importance of ethical considerations in data usage. Our study ensures that the handling of data excludes any sensitive or personal information. The use of data strictly adheres to academic standards for transparency and ethical practices concerning public datasets.
> >
> > ---

---

> > > ### Author Response · Authors · 2024-11-26
> > > **Sincerely Looking Forward to More Discussions**
> > >
> > > Dear Reviewer EAeh,
> > >
> > > Thank you for your valuable feedback. We have carefully addressed all your concerns and deeply appreciate your thoughtful suggestion to revise the manuscript.
> > >
> > > Regarding evaluation settings, we emphasized our resource-based decision to prioritize zero-shot LLM evaluations on challenging tasks while ensuring clarity in our comparative framework. For ethical concerns, we clarified that our dataset adheres to legal mandates for public disclosure and complies with all ethical standards. Additionally, we justified the exclusion of domain-specific financial LLMs by highlighting their suboptimal performance in our preliminary experiments. To reflect real-world conditions, we preserved the class imbalance in our dataset, ensuring robust evaluation of edge cases. We further clarified data legitimacy, ethical compliance, and academic-purpose usage, emphasizing proper attribution and non-commercial application to align with ethical and legal requirements.
> > >
> > > We hope these revisions comprehensively address your comments and are happy to discuss any additional concerns you may have.
> > >
> > > Best regards.

---

> > ### Author Response · Authors · 2024-12-02
> >
> > Dear Reviewer EAeh,
> >
> > Thank you for your valuable feedback. We have carefully addressed all your concerns and deeply appreciate your thoughtful suggestion to revise the manuscript.
> >
> > Regarding evaluation settings, we emphasized our resource-based decision to prioritize zero-shot LLM evaluations on challenging tasks while ensuring clarity in our comparative framework. For ethical concerns, we clarified that our dataset adheres to legal mandates for public disclosure and complies with all ethical standards. Additionally, we justified the exclusion of domain-specific financial LLMs by highlighting their suboptimal performance in our preliminary experiments. To reflect real-world conditions, we preserved the class imbalance in our dataset, ensuring robust evaluation of edge cases. We further clarified data legitimacy, ethical compliance, and academic-purpose usage, emphasizing proper attribution and non-commercial application to align with ethical and legal requirements.
> >
> > We hope these revisions comprehensively address your comments and are happy to discuss any additional concerns you may have.
> >
> > Best regards.

---

### Meta-Review · Area_Chair_jNx7 · 2024-12-22

**Metareview:**

This paper presents a new dataset for assessing the quality of responses on Chinese stock exchange investor platforms. In particular, the proposed FinTruthQA dataset includes 6,000 Q&A entries, each manually annotated across four evaluation criteria: question identification, question relevance, answer readability, and answer relevance. Although the new dataset would make a very meaningful contribution to the research community, reviewers found that there still major limitations in the current version, such as limited evaluation settings, definition of readability, label distribution, lack of in-depth analysis, etc. Although some of these issues have been addressed by the authors' rebuttal, during the post-rebuttal discussions, reviewers believed that some remaining issues (such as the class imbalance issue) cannot be addressed by a minor revision. Therefore, this paper in its current version is not ready for publication at ICLR.

**Additional Comments On Reviewer Discussion:**

Although this paper would make a very meaningful contribution to the research community by providing a new dataset on AI and Finance, reviewers agreed that there still remain major limitations, such as the class imbalance issue in the dataset.

---

### Decision · Program_Chairs · 2025-01-22

Reject